

# Comparative analysis of prophage-like elements in *Helicobacter* sp. genomes

Xiangyu Fan, Yumei Li, Rong He, Qiang Li and Wenxing He

School of Biological Science and Technology, University of Jinan, Jinan, China

## ABSTRACT

Prophages are regarded as one of the factors underlying bacterial virulence, genomic diversification, and fitness, and are ubiquitous in bacterial genomes. Information on *Helicobacter* sp. prophages remains scarce. In this study, sixteen prophages were identified and analyzed in detail. Eight of them are described for the first time. Based on a comparative genomic analysis, these sixteen prophages can be classified into four different clusters. Phylogenetic relationships of Cluster A *Helicobacter* prophages were investigated. Furthermore, genomes of *Helicobacter* prophages from Clusters B, C, and D were analyzed. Interestingly, some putative antibiotic resistance proteins and virulence factors were associated with *Helicobacter* prophages.

## INTRODUCTION

Prophages, a type of phage that integrates into and remains in a bacterial genome, play an important role in the genomic diversification and fitness cost of bacteria to the infected host. As a class of genetic elements, some prophages can mediate horizontal gene transfer in the evolution of bacterial genomes (*Lang, Zhaxybayeva & Beatty, 2012*). Because they carry virulence genes, some prophages make outstanding contributions to bacterial pathogenesis (*Penadés et al., 2015*) and some have also contributed to the fitness cost of bacteria to the infected host (*Fan et al., 2013*). Therefore, it is essential to search for the presence of prophages in the bacterial genomes and to analyze them. To date, studies have identified prophages in a diverse range of hosts, such as *Moraxella catarrhalis* (*Ariff et al., 2015*), *Lawsonia intracellularis* (*Vannucci, Kelley & Gebhart, 2013*), *Bifidobacterium* spp. (*Lugli et al., 2016*; *Ventura et al., 2009*), *Lactococcus* spp. (*Ventura et al., 2007*), *Mycobacterium* spp. (*Fan, Abd Alla & Xie, 2015*; *Fan et al., 2014*), *Streptococcus* spp. (*Tang et al., 2013*), and some plant-pathogenic bacteria (*Varani et al., 2013*). However, a systemic investigation of genomic information and function of *Helicobacter* prophages is largely lacking.

*Helicobacter* is a genus of Gram-negative bacteria, most frequently found in the upper gastrointestinal tract of mammals. One well-known species of the genus is *Helicobacter pylori*, a carcinogen identified by the World Health Organization (*Uemura et al., 2001*). *H. pylori* infection may be associated with gastritis, peptic ulcer, and gastric cancer (*Peek & Blaser, 2002*). Other non-*pylori Helicobacter* species such as *H. suis*, *H. felis*, *H. bizzozeronii* and *H. salomonis* have been reported and also exhibit carcinogenic potential in animals (*O'rourke, Grehan & Lee, 2001*). Previous research suggests that *Helicobacter* phages and

Corresponding authors
Qiang Li, lq_ujn@126.com
Wenxing He, wxh_ujn@126.com

prophages are unusual (*Canchaya, Fournous & Brüssow, 2004*). Information on *Helicobacter* prophages is becoming increasingly available. Two prophage-like elements were detected in *Helicobacter acinonychis str.* Sheeba (*Eppinger et al., 2006*). One prophage-like element was found within *Helicobacter felis* ATCC 49179 (*Arnold et al., 2011*). One prophage, phiHP33, which can be induced by UV irradiation, was found in *H. pylori* B45 (*Lehours et al., 2011*). Luo and colleagues (*2012*) found that the *H. pylori str.* HP1961 chromosome contains a full-length prophage 1961P. Luo also found that *H. pylori* Cuz20, *H. pylori* India7, *H. pylori* B38, *H. pylori* F16, and *H. pylori* Gambia94/24 chromosomes all contain a prophage-like element (*Luo et al., 2012*). In addition, two potential prophages were described in *H. pylori str.* Egypt (*Abdel-Haliem & Askora, 2013*). These findings suggest that prophages are common within the *Helicobacter* genomes. *Vale et al. (2015)* have demonstrated that prophages play a role in the diversity of *H. pylori*. The function of *Helicobacter* prophages is nonetheless ill-defined. Some researchers suggest that it is possible to use *Helicobacter* phages to control some diseases caused by *H. pylori* (*Abdel-Haliem & Askora, 2013*). However, if virulence factors and antibiotic resistance genes are found associated with *Helicobacter* phages or prophages, it is worth reconsidering phage therapy as treatment of *H. pylori* infections. As of 1 Oct 2015, eighty-one *Helicobacter* species genomes have been sequenced and assembled. These comprise an essential dataset for researching the presence of *Helicobacter* prophages.

As mentioned above, it is important that "hidden" *Helicobacter* prophages are identified. In this study, we screened all the available complete *Helicobacter* sp. genome sequences deposited in GenBank for the presence of prophages. We here report the results of our comparative genomic analysis, genome content analysis, and prophage-encoded virulence and antibiotic resistance gene analysis of *Helicobacter* prophages.

## MATERIALS AND METHODS

### Data collection and identification of *Helicobacter* prophages

Eighty-one complete *Helicobacter* genomes were downloaded from NCBI (the National Center for Biotechnology Information). *Helicobacter* prophages were identified using a previously reported method (*Fan et al., 2014*). In the first place, we used PHAST (http://phast.wishartlab.com/index.html) to analyze bacterial genomes to find candidate prophages. Next, we screened integrase gene from prophage genomes to drop false positives results. Finally, based on the presence of significant homology between ORFs (open reading frames) and known phage genes, we obtain *Helicobacter* prophages.

### Genomic and comparative genomic analyses of *Helicobacter* prophages

Prophage flanking sites *attL* and *attR* were identified using DNAMAN. Prophage genes were annotated using Glimmer (*Delcher et al., 2007*). Dot plot comparisons of *Helicobacter* prophage genomes were carried out with Geneious software (*Kearse et al., 2012*). Global genome comparison was performed using BLASTn, at NCBI (http://blast.ncbi.nlm.nih.gov/Blast.cgi), and results were shown by ACT software. For all software, default settings were used.

## RESULTS AND DISCUSSION

### Prophages in *Helicobacter* sp. genomes

Eighty-one complete *Helicobacter* sp. genomes (Table S1) were retrieved. Thirteen prohages (Table 1) were detected using a previously reported method (*Fan et al., 2014*), eight of them were novel, and five of them have been described in the literature (*Luo et al., 2012*). Moreover, seven reported prophages (Table 1) from *Helicobacter* genomes were not detected in the screen (*Arnold et al., 2011*; *Eppinger et al., 2006*; *Lehours et al., 2011*; *Luo et al., 2012*). Two of them, contained in the genomes of *H. acinonychis str.* Sheeba and *H. felis* ATCC 49179, have not been designated. We named them phiHac_1 and phiHFELIS_1, respectively. It is worth noting that phiHac_1, phiHFELIS_1 and two other prophages from *H. pylori str.* Egypt, ΦHPE1 and ΦHPE2, all lack sequence information. The original papers where these prophages were identified did not provide the sequence information and we cannot retrieve it from the corresponding genomes using our screening method. We therefore discarded them during follow-up analyses. In general, sixteen prophages are analysed.

The size of all *Helicobacter* prophage genomes varies between 5.5 kb and 39.3 kb. Based on the presence of predicted prophage proteins and the length of the prophage genomes, nine sequences were designated as full-length prophages, and seven sequences were labeled prophage-like elements.

### Comparative genomics of *Helicobacter* (pro)phages

We carried out a comparative genomics analysis of sixteen *Helicobacter* prophages with known sequence information using dot plot matrix (Fig. 1). Two *Helicobacter* phages, KHP30 (*Uchiyama et al., 2013*) and KHP40 (*Uchiyama et al., 2012*), were selected as the reference for DotPlots. This revealed that most *Helicobacter* (pro)phages can be sorted into a common group called a 'cluster' (designated 'Cluster A') based on the similarities of their genomes. *Helicobacter* phages of Cluster A can be further divided into subclusters, according to their genomic sequences. These were designated subcluster A1 (containing phiNY40_1, phiK750_1, Sheeba, KHP30, KHP40, 1961P, phiHP33, Cuz20 and India7), subcluster A2 (containing Gambia94/24, phiK747_1, phiK749_1 and phiK748_1), subcluster A3 (B38), and subcluster A4 (F16), respectively. Other *Helicobacter* phages were grouped into Cluster B (phiHH_1), Cluster C (phiHCD_1), and Cluster D (phiHBZC1_1), as appropriate.

### *Helicobacter* phage Cluster A

Based on the similarities of their genomes, *Helicobacter* Cluster A phages were divided into four subclusters. Phages belonging to one subcluster are more closely related to each other than to phages in the remaining subclusters (Figs. S1 and S2). Some subcluster A1 phages (phiK750_1, Sheeba, KHP30, KHP40, 1961P, phiHP33, Cuz20 and India7) possess 70.57% identity with each other, as determined by multiple genomic sequence alignments in DNAMAN. In addition, a BLASTn comparison of phiNY40_1 and phiK750_1 revealed one major sequence segment (8,953 bp) with 81% identity and three segments (3,550 bp, 3,039 bp, and 1,997 bp) with identity greater than 76%. Based on the multiple genomic sequence alignments, all subcluster A2 phages displayed 82.79% identity between each other.

Fan et al. (2016), *PeerJ*, DOI 10.7717/peerj.2012

**Table 1** Genomic features of prophages in *Helicobacter* genomes.

| Prophages | Cluster | Host | Accession numbers of bacteria | Coordinates | Size | Putative *att* B regions of prophage-like elements | References |
|---|---|---|---|---|---|---|---|
| phiK747_1[a] | Cluster A2 | *Helicobacter pylori* UM032 | CP005490.3 | 1500592–1515028 | 14.4 kb | AAACAAATTTTTAAAA | this study |
| phiK749_1[a] | Cluster A2 | *Helicobacter pylori* UM299 | CP005491.3 | 487627–502064 | 14.4 kb | AAACAAATTTTTAAAA | this study |
| phiK750_1[a] | Cluster A1 | *Helicobacter pylori* UM037 | CP005492.3 | 1184664–1213258 | 28.6 kb[d] | ATTGATAGAAATAAT | this study |
| phiK748_1[a] | Cluster A2 | *Helicobacter pylori* UM298 | CP006610.2 | 167091–181528 | 14.4 kb | AAACAAATTTTTAAAA | this study |
| phiNY40_1[a] | Cluster A1 | *Helicobacter pylori* NY40 | AP014523.1 | 523881–555620 | 31.7 kb[d] | TTTTTGTGATTGAT | this study |
| phiHH_1[a] | Cluster B | *Helicobacter hepaticus* ATCC 51449 | AE017125.1 | 732167–748393 | 16.2 kb | AATCAAAGTGAGAGA | this study |
| phiHCD_1[a] | Cluster C | *Helicobacter cetorum* MIT 99-5656 | CP003481.1 | 178240–203078 | 24.8 kb[d] | AAACACTTTTAAA | this study |
| phiHBZC1_1[a] | Cluster D | *Helicobacter bizzozeronii* CIII-1 | FR871757.1 | 1613405–1669733 | 39.3 kb[d] | CTTTATCAAAATGC | this study |
| Cuz20[ab] | Cluster A1 | *Helicobacter pylori* Cuz20 | CP002076.1 | 186400–215514 | 29.1 kb[d] | TTATAGCTTATTTCA | (*Luo et al., 2012*) |
| India7[ab] | Cluster A1 | *Helicobacter pylori* India7 | CP002331.1 | 1217797–1246918 | 29.1 kb[d] | TTATAGCTTATTTCA | (*Luo et al., 2012*) |
| B38[ab] | Cluster A3 | *Helicobacter pylori* B38 | FM991728.1 | 1513448–1518986 | 5.5 kb | TTATAG (*attL*)[e] | (*Luo et al., 2012*) |
| Gambia94/24[ab] | Cluster A2 | *Helicobacter pylori* Gambia94/24 | CP002332.1 | 202163–218412 | 16.3 kb | TTATAGCTAATT (*attL*) TTATAGCTTATTTCA (*attR*) | (*Luo et al., 2012*) |
| phiHac_1[bc] | c | *Helicobacter acinonychis* str. Sheeba | AM260522.1 | NM | 11.6 kb | NM | (*Eppinger et al., 2006*) |
| Sheeba[ab] | Cluster A1 | *Helicobacter acinonychis* str. Sheeba | AM260522.1 | 1396699–1425613 | 28.9 kb[d] | AAGATATCTCTTATT | (*Eppinger et al., 2006*) |
| F16[b] | Cluster A4 | *Helicobacter pylori* F16 | AP011940.1 | 470905–485827 | 14.9 kb | TTATAGCTTATTTCA (*attL*)[e] | (*Luo et al., 2012*) |
| phiHP33 (B45)[b] | Cluster A1 | *Helicobacter pylori* B45 | JF734911.1 | NM | 24.6 kb[d] | TTATAGCTTATTTCA (*attL*) TTATAGCTTATTT (*attR*) | (*Lehours et al., 2011*) |
| 1961P[b] | Cluster A1 | *Helicobacter pylori* strain HP1961 | Not found | NM | 26.8 kb[d] | TTATCTTT | (*Luo et al., 2012*) |
| phiHFELIS_1[bc] | c | *Helicobacter felis* ATCC 49179 | FQ670179.2 | NM | NM | NM | (*Arnold et al., 2011*) |
| ΦHPE1[bc] | c | *Helicobacter pylori* str. Egypt | Not found | NM | NM | NM | (*Abdel-Haliem & Askora, 2013*) |
| ΦHPE2[bc] | c | *Helicobacter pylori* str. Egypt | Not found | NM | NM | NM | (*Abdel-Haliem & Askora, 2013*) |

**Notes.**

NM means that these data were not mentioned.

[a] Those prophages were detected in the screen.

[b] Those prophages had been described in the literature.

[c] The prophage lack sequence information.

[d] Those prophages are full-length prophage.

[e] Absent *attR* from the junction.
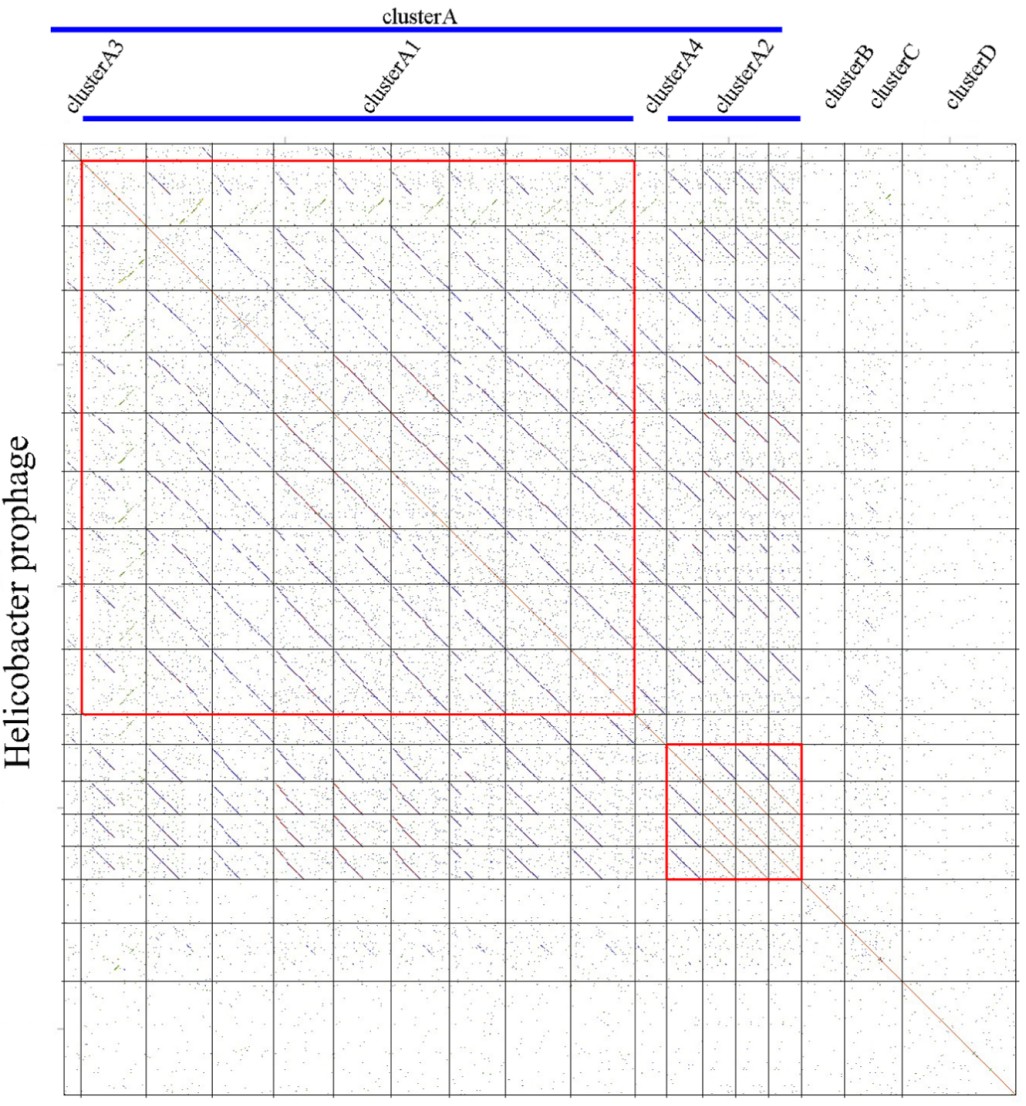

**Figure 1** **Comparative genomic analyses of *Helicobacter* prophages.** There are 16 *Helicobacter* prophages and 2 *Helicobacter* phages. The order of phages was B38, phiNY40_1, phiK750_1, Sheeba, KHP30, KHP40, 1961P, phiHP33, Cuz20, India7, F16, Gambia94/24, phiK747_1, phiK749_1, phiK748_1, phiHH_1, phiHCD_1 and phiHBZC1_1. The clusters of related phages (Clusters A, B, C and D) are shown in the figure. Geneious software was used to carry out dot plot analysis. The word length used is 13 bp.

Different subclusters in *Helicobacter* phage Cluster A possess segments of DNA similarity. Phages of subclusters A2, A3, and A4 all shared sequence similarity with subcluster A1 phages (Fig. 2). These are remnant prophage-like elements that have lost sequence segments during evolution. Subcluster A2 prophages retained an upstream region with many virion-associated genes of the subcluster A1 prophages. Subcluster A3 prophage (prophage B38) retained only an incomplete upstream region (5.5 kb) of subclusters A1 and A2 prophages. Subcluster A4 prophage (prophage F16) retained a downstream region containing many

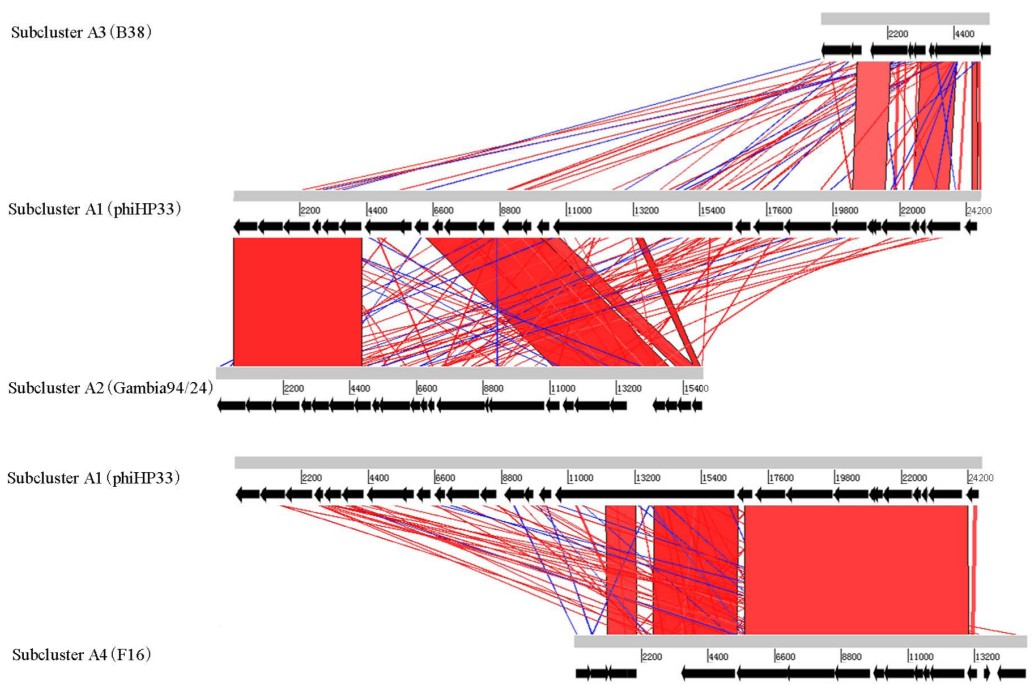

**Figure 2 Global comparison of representative phages of Cluster A.** The red shading means that the fragments are homologous to other fragments. The results were obtained by Blast-N and depicted by ACT software. Numbers indicate the length of genomes (bp).

DNA metabolism genes of the subcluster A1 prophages. Genome organization of most Cluster A phages has been reported (*Luo et al., 2012*).

### *Helicobacter* phage Cluster B

Cluster B contains only one *Helicobacter* prophage, phiHH_1. The genome size of phiHH_1, which lacks the lysin gene, is 16.2 kb. Therefore, phiHH_1 is considered to be a prophage-like element. This prophage is integrated into the *H. hepaticus* ATCC 51,449 genome, extends from HH_0750 (the integrase gene) to HH_0772 (encoding a carbohydrate-binding protein), and contains twenty-three ORFs (Fig. 3; Table S2). PhiHH_1 prophage is flanked by 15 bp *attL* and *attR* sites (Table 1). Twelve ORFs were assigned phage gene status after homologous analysis of protein sequences (Table S2). Based on database searches, nine of these encode specific functions, namely, integrase (HH_0750), DNA transposition protein (HH_0752), host-nuclease inhibitor protein Gam (HH_0754), Rha family transcriptional regulator (HH_0755), DNA-binding protein RdgB (HH_0756), phage Tail Collar Domain family (HH_0761), DNA methyltransferase (HH_0763), tape measure protein (HH_0771), and carbohydrate-binding protein (HH_0772).

### *Helicobacter* phage Cluster C

Although *Helicobacter* prophage phiHCD_1 displays some similarity to the subcluster A1 and A4 phages, it is not sufficiently closely related to be included in a common cluster. Therefore, phiHCD_1 is categorized into Cluster C. The genome size of phiHCD_1 is 24.8 kb, which renders it a full-length prophage. Prophage phiHCD_1, inserted between

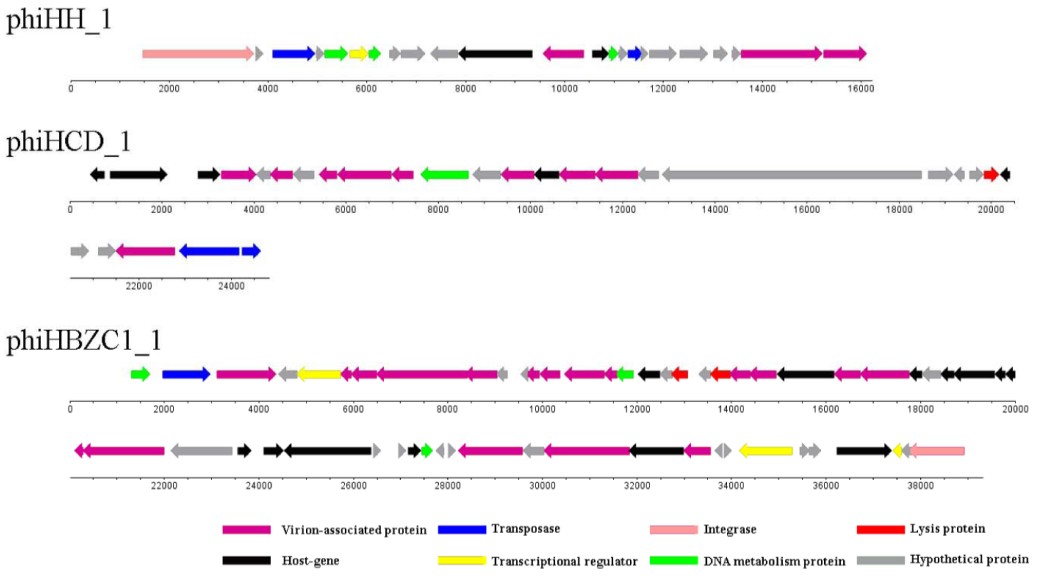

**Figure 3** The genomic organization of *Helicobacter* prophage phiHH_1, phiHCD_1 and phiHBZC1_1. *Helicobacter* prophage genes are grouped into eight functional modules: lysis module, DNA packaging and virion-associated modules, DNA metabolism module, transcriptional regulatory module, lysogeny module, host protein module and hypothetical protein module. The functions of the proteins are displayed by color coding. Dnaplotter software was used to draw the figure. Numbers indicate the length of genomes (bp).

HCD_00885 (thioredoxin-encoding) and HCD_01020 (transposase-encoding) in the genome of *Helicobacter cetorum* MIT 99-5656, contains twenty-eight ORFs (Fig. 3). The prophage has identical 13 bp *attL* and *attR* sites (Table 1). Based on amino acid sequence homology, we identified eighteen ORFs that have sequence similarity to genes of other phages. It was possible to assign function to thirteen of them (Table S3). These are, accordingly: terminase (HCD_00900); phage tail tape measure protein (HCD_00910); phage structure protein (HCD_00920, HCD_00930, and HCD_00960); phage major capsid protein (HCD_00925); UV radiation resistance protein (HCD_00935); phage prohead protease (HCD_00945); phage tail protein (HCD_00955); holin (HCD_00990); portal protein (HCD_01010); transposase (HCD_01015, and HCD_01020).

### *Helicobacter* phage Cluster D

PhiHBZC1_1 is found in *Helicobacter bizzozeronii* CIII-1. It belongs to Cluster D and does not share any similarities with other *Helicobacter* phages. As a full-length prophage, the genome size of phiHBZC1_1 is 39.3 kb. There are fifty-eight ORFs in this genome (Fig. 3), spanning a region from HBZC1_17420 (DNA invertase-encoding) to HBZC1_17990 (site-specific recombinase integrase-encoding). The prophage is flanked by two 14 bp *attL* and *attR* sites (Table 1). Sequence alignment analysis indicated some level of similarity between thirty ORFs of prophage phiHBZC1_1 and other known phage genes. Of these, twenty-eight ORFs could be assigned biological functionalities (Table S4).

The genome of phiHBZC1_1 can be divided into several different functional modules. The lysis module includes HBZC1_17600 and HBZC1_17620, which encode a holin and a lysozyme protein, respectively. The DNA packaging and virion-associated modules

consist of HBZC1_17440, coding for a phage terminase large subunit; HBZC1_17470, encoding a phage tail protein; phage tail tape measure proteins-encoding HBZC1_17480, HBZC1_17490, and HBZC1_17500; phage tail proteins-encoding HBZC1_17530, HBZC1_17540, HBZC1_17550, HBZC1_17630, HBZC1_17640, and HBZC1_17660; HBZC1_17560, encoding a phage tail sheath-like protein; HBZC1_17670, encoding a phage baseplate protein; capsid proteins-encoding HBZC1_17740 and HBZC1_17750; HBZC1_17860, encoding a portal protein; HBZC1_17880, encoding a phage terminase large subunit; and HBZC1_17900, encoding a phage baseplate assembly protein V. The DNA metabolism module comprises of three genes (HBZC1_17420, HBZC1_17570, and HBZC1_17830), whose predicted protein products are phage DNA invertase, DNA methyltransferase, and DNA polymerase, respectively. The transcriptional regulatory module is composed of HBZC1_17460 (encoding a phage late control D family protein), HBZC1_17930 (coding for the repressor LexA), and HBZC1_17970 (encoding a YcfA family protein). The lysogeny module appears to be limited to HBZC1_17990, whose predicted protein product is a phage integrase.

### Putative antibiotic resistance genes and virulence factors associated with *Helicobacter* prophages

Except for phiHBZC1_1, none of the other characterized *Helicobacter* prophages contain known antibiotic resistance genes. The protein encoded by HBZC1_17700 shows high similarity to multidrug resistance protein D (emrD) of *Salmonella enterica* subsp. enterica serovar Infantis (Table 2). Multidrug resistance protein D belonging to the major facilitator superfamily facilitates the transport of a variety of antibiotics (*Shaheen et al., 2015*).

A range of phage-encoded virulence genes was identified within the *Helicobacter* prophage sequences (Table 2). A DNA methyltransferase-encoding gene was identified in most of the analyzed *Helicobacter* prophages. DNA methyltransferase is thought to contribute to the specificity of bacterium-host interactions or *H. pylori* virulence (*Vitkute et al., 2001*). Furuta and colleagues (*2015*) found that DNA methyltransferase genes are rapidly evolving in *H. pylori* genomes, which facilitates *H. pylori* adaptation to a new host. A protein encoded by phiNY40_1 (NY40_0553) displayed 23% identity with a serine/threonine kinase of *Thiorhodococcus drewsii*. Phosphorylation of proteins usually occurs during interactions between bacterial cells and host cells and plays a role in bacterial pathogenesis (*Cozzone, 2005*). Serine/threonine kinases are considered to affect cell survival pathways and contribute to *H. pylori* pathogenesis (*King & Obonyo, 2015*). A putative glycosyltransferase is encoded by phiHCD_1. Glycosyltransferases are involved in biosynthesis of LPS (*Luke et al., 2010*) that can promote proliferation of gastric cancer cells (*Tomoda, Kamiya & Suzuki, 2015*). An antitoxin component RelB of the addiction toxin-antitoxin (TA) module system RelBE was identified in phiHBZC1_1. The protein plays a role in cell survival (*Park, Son & Lee, 2013*).

## CONCLUSIONS

In brief, we present here sixteen *Helicobacter* prophages. Eight of them were identified for the first time after mining the sequenced *Helicobacter* sp. genomes, and the other eight had

**Table 2  Putative virulence elements and antibiotic resistance genes in *Helicobacter* prophages.**

| Prophage | Gene (Accession number) | Putative virulence element | Query coverage | *E*-value | Identity |
|---|---|---|---|---|---|
| KHP40 | ORF24 (BAM34796.1) | DNA methyltransferase (*Helicobacter pylori*) | 100% | 8e–41 | 91% |
| KHP30 | ORF23 (BAM34765.1) | DNA methyltransferase (*Helicobacter pylori*) | 100% | 1e–40 | 92% |
| 1961P | gp26 (AFC61925.1) | DNA methyltransferase (*Helicobacter pylori*) | 100% | 6e–44 | 96% |
| Cuz20 | HPCU_00990 (ADO03382.1) | DNA methyltransferase (*Helicobacter pylori*) | 100% | 2e–42 | 100% |
| India7 | HPIN_06120 (ADU80418.1) | DNA methyltransferase (*Helicobacter pylori*) | 100% | 2e–47 | 100% |
| Gambia94/24 | HPGAM_01040 (ADU81058.1) | DNA methyltransferase (*Helicobacter pylori*) | 100% | 1e–45 | 100% |
| phiK747_1 | K747_07685 (AGL67312.1) | DNA methyltransferase (*Helicobacter pylori*) | 100% | 2e–41 | 92% |
| phiK749_1 | K749_02305 (AGL67850.1) | DNA methyltransferase (*Helicobacter pylori*) | 100% | 2e–41 | 92% |
| phiK750_1 | K750_05880 (AGL70120.1) | DNA methyltransferase (*Helicobacter pylori*) | 95% | 4e–37 | 87% |
| phiK748_1 | K748_00765 (AGR63209.1) | DNA methyltransferase (*Helicobacter pylori*) | 100% | 2e–41 | 92% |
| phiNY40_1 | NY40_0558 (BAO97577.1) | Type II methylase (*Helicobacter pylori*) | 100% | 0.0 | 100% |
| phiNY40_1 | NY40_0553 (BAO97572.1) | Serine/threonine protein kinase (*Thiorhodococcus drewsii*) | 99% | 6e–56 | 23% |
| phiNY40_1 | NY40_0545 (BAO97564.1) | DNA methyltransferase (*Helicobacter pylori*) | 100% | 2e–43 | 100% |
| phiHH_1 | HH_0763 (AAP77360.1) | DNA methyltransferase (*Helicobacter sp.* MIT 03-1614) | 84% | 3e–31 | 97% |
| Sheeba | Hac_1629 (CAK00337.1) | DNA methyltransferase (*Helicobacter pylori*) | 97% | 1e–29 | 69% |
| phiHCD_1 | HCD_00890 (AFI05210.1) | Glycosyltransferase (*Neisseria meningitidis*) | 71% | 3e–71 | 17% |
| phiHBZC1_1 | HBZC1_17570 (CCB80743.1) | DNA methyltransferase (*Helicobacter sp.* MIT 03-1614) | 42% | 4e–08 | 55% |
| phiHBZC1_1 | HBZC1_17680 (CCB80754.1) | Type VI secretion protein (*Herbaspirillum sp.* B39) | 64% | 2.4 | 28% |
| phiHBZC1_1 | HBZC1_17710 (CCB80757.1) | DNA methyltransferase (*Oceanospirillum beijerinckii*) | 89% | 3e–66 | 39% |
| phiHBZC1_1 | HBZC1_17820 (CCB80754) | Addiction module antitoxin RelB (*Burkholderia cenocepacia*) | 91% | 2e–24 | 53% |
| phiHBZC1_1 | HBZC1_17770 (CCB80763.1) | DNA adenine methylase (*Campylobacter jejuni* subsp. jejuni 2008-979) | 89% | 6e–19 | 39% |
| phiHBZC1_1 | HBZC1_17780 (CCB80764.1) | DNA adenine methylase (*Desulfosporosinus acidiphilus*) | 87% | 2e–25 | 44% |
| phiHBZC1_1 | HBZC1_17700 (CCB80756.1) | Multidrug resistance protein D (*Salmonella enterica* subsp. enterica serovar Infantis) | 40% | 2e–06 | 31% |

been reported in published literature. Based on comparative genomic analyses, the sixteen phages were sorted into four clusters, Clusters A–D, respectively. Cluster A was further divided into four subclusters, subclusters A1–A4. Different subclusters displayed similarity to each other. Subcluster A1 phages are full-length prophages. Subcluster A2, A3 and A4 phages are remnant prophage-like elements. The genomes and genetic information of the Cluster B, C and D phages were analyzed. Interestingly, several genes encoding antibiotic resistance proteins and virulence factors were found within various prophage genomes. These results highlight an important issue, which needs to be resolved before proceeding with phage therapy for treatment of *H. pylori* infections. To our knowledge, this is the first systematic analysis of *Helicobacter* prophages. With more forthcoming *Helicobacter* genome sequences, more *Helicobacter* prophages will be identified, and the role of prophages in evolution, adaptations and physiology of *Helicobacter* sp. will be clarified.

### Funding

This work was supported by Shandong Excellent Young Scientist Award Fund (BS2014YY031), Foundation of University of Jinan (XBS1519, XKY1324), National Natural Science Foundation of China (31100088, 31300045, 51208290, 31372356), Shandong province science and technology development plan (2013GSF12006), A Project of Shandong Province Higher Educational Science and Technology Program (YE13), Open Foundation of Xinjiang Production & Construction Corps Key Laboratory of Protection and Utilization of Biological Resources in Tarim Basinand (BYBR1405, BRYB1501). The funders had no role in study design, data collection and analysis, decision to publish, or preparation of the manuscript.

### Grant Disclosures

The following grant information was disclosed by the authors:
Shandong Excellent Young Scientist Award Fund: BS2014YY031.
Foundation of University of Jinan: XBS1519, XKY1324.
National Natural Science Foundation of China: 31100088, 31300045, 51208290, 31372356.
Shandong province science and technology development plan: 2013GSF12006.
Project of Shandong Province Higher Educational Science and Technology Program: YE13.
Open Foundation of Xinjiang Production & Construction Corps Key Laboratory of Protection and Utilization of Biological Resources in Tarim Basinand: BYBR1405, BRYB1501.

### Competing Interests

The authors declare there are no competing interests.

### Author Contributions

- Xiangyu Fan conceived and designed the experiments, performed the experiments, analyzed the data, wrote the paper, prepared figures and/or tables.
- Yumei Li and Rong He reviewed drafts of the paper.
- Qiang Li and Wenxing He contributed reagents/materials/analysis tools, reviewed drafts of the paper.

### Data Availability

Raw data were provided as Supplemental Information.

### Supplemental Information

Supplemental information for this article can be found online at http://dx.doi.org/10.7717/peerj.2012#supplemental-information.

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
