# Peer review of "Comparative analysis of prophage-like elements in Helicobacter sp. genomes"

_PeerJ, doi:10.7717/peerj.2012_

## Round 0.1 · original submission · Major Revisions

Please address all of the reviewer's comments, especially those related to developing the methods section, clarifying the first paragraph of the results and providing more information in the figure legends. In addition, I recommend to add a schematic figure comparing the structure of the incomplete prophages in subclusters A2, A3 and A4 to that of the complete prophages in subcluster A1.

Also, please address the following:

-in the introduction, clarify what is meant by "fitness cost of bacteria". Is this referring to the fitness cost of bacteria to the infected host? or do you mean simply "fitness of bacteria"?
-in lines 92-93 of the results, explain the "lack of sequence information" for the mentioned prophages. Don't the original papers where these prophages were identified provide the sequence information or the means to retrieve it from the corresponding genomes?
-in the notes to Table 1, correct "do not be" and "did not be " to "were not"

Reviewer 1 ·

Basic reporting

Lines 85-93: It would be interesting to know if the authors were able to identify all 20 prophages using their screening method. Accordingly, pls restructure the first paragraph (81 were screened, xx prohages were detected, 8 were novel, 12 are described in the literature, xx were not detected in the screen – 2 according to Table 1?)

Cluster descriptions/Prophage gene description: the listings of genes and putative functions is hard to read, a table would provide a better overview; I could imagine a supplemental Table for each of the described prophages similar to Table 2 would be helpful and then highlight only the most relevant features in the text.

Line 186: pls provide more information/discussion about the multidrug resistance protein D and its relevance/function

Line 201: pls provide a Reference for your statement “promote proliferation of gastric cancer cells”

Line 99: Did the authors select KPH30 and KHP40 as the reference for DotPlots?

Table 1: It might be worthwhile to extent Table 1 with Hp genome accession numbers and to which cluster each prophage was assigned. Moreover, references for phiHP33 and following are missing.

Figure legeds are not sufficient. Pls indicate which programs were used, what colors, numbers,… mean.

Minor
Lines 27-28: I suggest removing the last sentence of the abstract.
Lines 33-34: I suggest writing the sentence in plural.
Line 70: I suggest removing: “We performed a Helicobacter literature search.”

Experimental design

Fan et al. screened 81 Helicobacter strains for the presence of prophages and extended their list with prophages previously reported in the literature. Furthermore, they classified these prophages and described some of them in more detail. Their work is of importance for the field and provides and good overview and a classification scheme. However, the manuscript is hard to follow due to extensive listings of prophage features. Above pls find some suggestions, which could help to improve the manuscript.

Methods
Figure 2 and Figure S2: Did the authors use Artemis to generate these figures? Pls state also in the Method section.
Pls indicate for the programs which settings were used (default?).

Validity of the findings

no comments

Reviewer 2 ·

Basic reporting

This manuscript describes a comparative study of prophage like elements in Helicobacter genomes:

Introduction:
1) The authors only give information of Helicobacter pylori while they also included genomes from other Helicobacter spp. Knowledge of other Helicobacter spp. should be mentioned as well.

2) Why did the authors only included complete genomes? There are draft genomes available from other gastric and enterohepatic species and would be very interesting to look for the presence of prophages in these genomes as well.

Material and method section: this is very short and should be extend a bit. They also do not refer to table 1 which includes the genomes analysed in this manuscript.

Results and Discusion:
section Prophages in Helicobacter sp. genomes: this is very unstructured and difficult to follow. How many prophages are analysed, how many were known and how many were newly identified?? This section should be rewritten.

section comparative genomics of Helicobacter prophages: the terms phages and prophages are used here and this terminology is very confusing.

Line 110-203: I advise to present the gene content of the prophages in figures as in figure 3. This will reduce the length of the results and discussion section and only the essential part discussed.

section: Conclusion: this lacks future perpectives

Experimental design

The research question is defined but the methodology is not well explained. This should be expanded so that the reader can follow.

Validity of the findings

Line 110-203: I advise to present the gene content of the prophages in figures as in figure 3. This will largely reduce the length of the results and discussion section and only the essential part discussed. Now this part is difficult to follow.

Conclusion part: this lacks future perpectives

---

## Round 0.2 · Minor Revisions

Please correct the wording indicated by the reviewer. In addition, although the Methods section has been developed in this version, some further clarification is needed:

-in lines 81-82, you state "we screened integrase gene from prophage genomes to drop false negative results". Do you mean that you searched for the integrase gene in the previously identified candidate prophages? In that case, one would expect that you would drop false positives (when such gene was not found) rather than false negatives. Please clarify.

Regarding figures, in Figure 1 please clarify why the numbers corresponding to genome length go up to 400 000 bp when the prophages are only up to 40 000 bp long.

Also, please do the following corrections:

-in line 39, change Bifidobacteria spp. to Bifidobacterium spp.
-in line 80, add an "s" to genome, and in line 81 replace "In the next place" by "Next"
-in line 89, change "showed" to "shown"
-in line 96, change "harboring" to "from"
-in line 162, replace "there" by "these"
-in line 230, replace "insight" by "clarified".
-in line 334, add an "s" to "cluster"
-in line 337, remove "s" from "shadings"
-in line 338, replace "carried out" by "obtained"
-in line 339, replace "shown" by "depicted"
-in line 344, replace "displaied" by "displayed"
-in Figure 2, add numbers corresponding to genome location
-in table 1, complete the empty columns for ΦHPE2 and add an "s" to the column header "Coordinate"

Reviewer 1 ·

Basic reporting

all my previous comments have been addressed appropriately

minor: pls change in figure legens "numbers means" to "numbers indicate"

Experimental design

no comments

Validity of the findings

no comments

---

## Round 0.3 · accepted · Accept

Thanks for complying with the suggested revisions.